# Structural Basis of Zika Virus Specific Neutralization in Subsequent Flavivirus Infections

**DOI:** 10.3390/v12121346

**Published:** 2020-11-24

**Authors:** Madhumati Sevvana, Thomas F. Rogers, Andrew S. Miller, Feng Long, Thomas Klose, Nathan Beutler, Yen-Chung Lai, Mara Parren, Laura M. Walker, Geeta Buda, Dennis R. Burton, Michael G. Rossmann, Richard J. Kuhn

**Affiliations:** 1Department of Biological Sciences, Purdue University, West Lafayette, IN 47907, USA; msevvana@purdue.edu (M.S.); mille153@purdue.edu (A.S.M.); longfe@whu.edu.cn (F.L.); tklose@purdue.edu (T.K.); gbuda@purdue.edu (G.B.); mr@purdue.edu (M.G.R.); 2Department of Immunology and Microbiology, The Scripps Research Institute, La Jolla, CA 92037, USA; trogers@scripps.edu (T.F.R.); nbeutler@scripps.edu (N.B.); yclai@scripps.edu (Y.-C.L.); mparren@scripps.edu (M.P.); burton@scripps.edu (D.R.B.); 3Adimab LLC, Lebanon, NH 03766, USA; laura.walker@adimab.com; 4Ragon Institute of Massachusetts General Hospital, Massachusetts Institute of Technology and Harvard University, Cambridge, MA 02139, USA; 5Purdue Institute of Inflammation, Immunology, and Infectious Disease, Purdue University, West Lafayette, IN 47907, USA

**Keywords:** secondary flavivirus infection, Zika antibody structure, Zika–dengue co-infection, flavivirus neutralization

## Abstract

Zika virus (ZIKV), a mosquito-borne human flavivirus that causes microcephaly and other neurological disorders, has been a recent focus for the development of flavivirus vaccines and therapeutics. We report here a 4.0 Å resolution structure of the mature ZIKV in complex with ADI-30056, a ZIKV-specific human monoclonal antibody (hMAb) isolated from a ZIKV infected donor with a prior dengue virus infection. The structure shows that the hMAb interactions span across the E protein dimers on the virus surface, inhibiting conformational changes required for the formation of infectious fusogenic trimers similar to the hMAb, ZIKV-117. Structure-based functional analysis, and structure and sequence comparisons, identified ZIKV residues essential for neutralization and crucial for the evolution of highly potent E protein crosslinking Abs in ZIKV. Thus, this epitope, ZIKV’s “Achilles heel”, defined by the contacts between ZIKV and ADI-30056, could be a suitable target for the design of therapeutic antibodies.

## 1. Introduction

Zika virus (ZIKV) is a mosquito-borne human pathogen and a member of the *Flaviviridae* family [1], closely related to dengue virus (DENV) [2], yellow fever virus (YFV) [3], West Nile virus (WNV) [4], Japanese encephalitis virus (JEV) [5] and tick-borne encephalitis virus (TBEV) [6]. ZIKV is the most widely studied flaviviruses after DENV, as a result of a major ZIKV epidemic in Brazil in 2015 with almost a million suspected cases [7,8]. It can cause congenital Zika syndrome in infants and Guillain-Barré syndrome in adults [9,10]. Development of an effective ZIKV vaccine and antiviral therapeutics are necessary to combat any future mass epidemics. Among several antiviral therapeutic strategies, neutralizing antibodies (nAbs) play a major role in protection against infection by flaviviruses [11,12,13]. High sequence conservation among flaviviruses causes immunological cross-reactivity. These cross-reactive nAbs, under sub-neutralizing concentrations, precipitate a severe disease phenomenon termed antibody-dependent enhancement of infection (ADE) [14,15,16,17,18]. Thus, understanding the immune response towards consecutive flavivirus infections and the mechanism of virus neutralization by various classes of nAbs may help to prevent disease severities leading to ADE [19,20].

Similar to other flaviviruses, ZIKV is an enveloped, single-stranded, positive-sense RNA virus. The 11 kb RNA genome is translated into a long polyprotein. It is post-translationally cleaved by host and viral proteases into three structural proteins: pre-membrane (prM), envelope (E) and the capsid (C), and seven non-structural proteins [21,22]. The E, prM and C proteins form a protective coat around the genome. The E protein mediates the assembly of virus, virus entry and fusion with host membrane, contains putative receptor binding sites and is a major target for nAbs [11,23,24,25,26]. The E protein forms a complex with prM in the endoplasmic reticulum shortly after its synthesis. The E-prM complex is arranged as 60 trimeric spikes on the surface of the immature virus, where the pr domain prevents premature fusion to host membranes [27]. Immature virions undergo pH-induced conformational changes in the trans-Golgi network followed by the cleavage of pr domain by furin to form mature particles [28]. The icosahedral mature ZIKV consists of 180 copies each of E and M proteins arranged in 60 asymmetric units [28,29] (Figure 1). Each icosahedral asymmetric unit consists of three E–M oligomers. Two E–M oligomers form a E–M heterodimer. Two adjacent asymmetric units, consisting of three E–M hetero-dimers, form a “raft”. Therefore, there are 30 rafts arranged in a herringbone pattern on the surface of the mature virus (Figure 1). Here, the E proteins near the two-fold, three-fold and five-fold axis of symmetry are named E2, E3 and E5, respectively, for convenience. There are two kinds of E protein dimers, E3-E5 and E2-E2’, that interact with nAbs, where E2’ is from an adjacent asymmetric unit. The nAb interacting sites near the two-fold, three-fold and five-fold axes of symmetry are described as 2f, 3f and 5f sites, respectively.

The ZIKV E ectodomain, which is the most exposed antigenic protein to nAbs, consists of three subdomains: a central ß-barrel domain I (E-DI), the dimerization domain II (E-DII), and an Ig-like binding domain III (E-DIII) [29,30] (Figure 1). The E-DII contains a highly conserved hydrophobic sequence, called the fusion loop (FL) required for virus–host membrane fusion. Flavivirus membrane fusion is initiated by receptor mediated endocytosis of the virus followed by low pH-induced conformational changes in the endosome, and exposure and interaction of the FL with the endosomal membrane. The E proteins rearrange into trimeric structures (fusogenic trimer) leading to the fusion of viral and endosomal membranes and subsequent release of the viral genome into the host cytosol [31,32,33]. NAbs might inhibit several of these virus entry and fusion processes during infection by inhibiting receptor binding, blocking conformational changes and exposure of the fusion loop.

The binding epitopes of nAbs against ZIKV are diverse. They are mostly distributed among E-DII and E-DIII with a few overlapping epitope residues on E-DI [20]. Several antigenic loops connecting secondary structure elements of E-DI, E-DII and E-DIII on the surface of the virus are shown in Figure 1. The mechanism of several ZIKV nAbs have been characterized and their epitopes mapped using structural studies [12,17,20,34,35,36,37,38,39,40,41,42,43,44]. Potently neutralizing ZIKV Abs either bind to epitopes on E protein domains (E-DI, E-DII and E-DIII) or to quaternary epitopes constituting more than one domain from neighboring E proteins within a dimer or in the interface between the two dimers. For example, both nAb Z006 (DENV cross-reactive) [35] and ZV-67 [36] are E-DIII specific, bind within a single E monomer and block flavivirus attachment and entry. The nAb Z3L [37] associates with regions across E-DI, E-DII and the hinge connecting them and blocks the conformational changes required for fusion. The E dimer dependent antibodies can be classified into two classes. The first class binds the interface between two E monomers within a dimer, for example the nAbs C8 (DENV cross-reactive) [17], A11 (DENV cross-reactive) [17], and Z20 (weakly cross-reactive against DENV) [37]. The second class, where the nAb epitope is spread across two E protein dimers, is represented in ZIKV-117 [45], C10 [17] and Z23 [37]. These nAbs block the conformational changes required for virus membrane fusion and inhibit domain reorganization as well as the formation of fusogenic trimers. In comparison, nAbs recognizing the fusion loop, for example the murine nAb 2A10G6 [12,34], are highly cross-reactive and less efficient in neutralizing ZIKV infection. The epitope information gained from these structures pinpoints vulnerable sites specific to ZIKV on its surface.

The evolution of ZIKV-induced B cell responses in three DENV infected donors has been reported [46]. Five months post-ZIKV infection, the donors developed ZIKV-specific antibodies in addition to the already existing DENV cross-reactive antibodies. Epitope mapping experiments showed that about 20% of the ZIKV-specific nAbs bound to epitopes within E-DIII and 15% bound to epitopes within or proximal to the fusion loop. However, about 60% of the ZIKV-specific nAbs targeted epitopes overlapping the nAb, ADI-30056 with IC_50_ values between 1.0 and 10 ng/mL. Therefore, we determined the structure of ZIKV in complex with ADI-30056 antigen-binding fragment (Fab) (ADI-Fab30056) using cryo-Electron Microscopy (cryo-EM) at a resolution of 4.0 Å. Both structural and functional analyses were used to identify binding determinants, binding mechanism and the evolution of ZIKV specific neutralization by ADI-30056 and related nAbs.

## 2. Materials and Methods

### 2.1. Expression and Purification of ADI-30056 and ADI-30056 Fab for Biochemical Assays

Anti–ZIKV IgGs were expressed in *S. cerevisiae* cultures grown in 24-well plates, as described previously [47]. Fab fragments were generated by digesting the IgGs with papain for 2 h at 30 °C. The digestion was terminated by the addition of iodoacetamide, and the Fab and fragment crystallizable (Fc) mixtures were passed over protein A agarose to remove Fc fragments and undigested IgG. The flowthrough of the protein A resin was then passed over CaptureSelect IgG-CH1 affinity resin (Thermo Fisher Scientific, Waltham, MA, USA) and eluted with 200 mM acetic acid and 50 mM NaCl (pH 3.5) into one-eighth volume of 2M 4-(2-hydroxyethyl)-1-piperazineethanesulfonic acid (HEPES) (pH 8.0). Fab fragments were then buffer-exchanged into phosphate-buffered saline (PBS; pH 7.0).

### 2.2. Generation of ZIKV and DENV for Biochemical Assays

Vero cells were cultured in minimal essential medium (MEM) (Corning Cellgro, New York, NY, USA) containing 5% heat-inactivated fetal bovine serum (Gibco-Invitrogen, Waltham, MA, USA). Zika virus stocks were supplied by K. Anderson and included strains: Paraiba-KY559032.1, Uganda-KU955594.1, Cambodia-FSS13025, Rio-KU926309, Panama-PA259459, and Puerto Rico-PRABC. Vero cells were infected with each virus and maintained until 50% reduction in viability was observed, after which supernatant was collected, filtered, and stored at −80 °C.

### 2.3. Focus Reduction Neutralization Test (FRNT)

Infection media was prepared from Dulbecco’s minimum essential medium (DMEM) (Life Tech Invitrogen, Carlsbad, CA, USA) containing 2% heat-inactivated fetal bovine serum (FBS) (Gibco). Antibody was diluted serially in infection media and incubated for 1 h at 37 °C with virus. Virus was added at equal volume to antibody solution, and diluted in infection media to achieve 100 plaque forming units (PFU). After incubation, the antibody-virus mixture was added to 96 well flat bottom plates (Corning) containing monolayer Vero E6 (American Type Culture Collection, Manassas, VA, USA) cells, which were plated with 3 × 10^4^ cells per well the previous day. Plates were incubated for 1.5 h at 37 °C. After incubation, wells were overlaid with 1% methylcellulose in DMEM, 2% heat inactivated fetal bovine serum (FBS) and 1:100 HEPES (Life Tech Invitrogen). Plates were incubated at 37 °C in 5% CO_2_ for 40 h. After incubation, cells were fixed with 100 µL of 1% paraformaldehyde in PBS for 1 h at 37 °C. After fixation, the cells were washed three times with 200 µL of 1× PBS containing 0.05% Tween. Cells were then incubated with a 1:2000 dilution of anti-flavivirus antibody (MAB10216, EMD Millipore, St. Louis, MI, USA) in Perm/Wash buffer (BD Bioscience, San Jose, CA, USA) for 2 h. After washing, cells were incubated with a 1:2000 dilution of anti-mouse horseradish peroxidase conjugated secondary antibody (115035146, Jackson Immuno Research Laboratories, West Grove, PA, USA) in Perm/Wash buffer for 2 h. After washing, the cells were permeabilized with 50 µL of Perm/Wash buffer for 10 min, and then developed with True-blue peroxidase substrate (KPL Inc. Gaithersburg, MD, USA). Neutralization curves were generated using Graphpad Prism software.

### 2.4. Enzyme-Linked ImmunoSorbent Assay (ELISA)

For whole virus binding, 48 µg/mL of anti-flavivirus E monoclonal antibody (mAb), 4G2 in phosphate buffer saline (PBS, pH 7.3) was coated on ELISA plates at 4 °C overnight. After blocking with 3% bovine serum albumin (BSA) in PBS for 1.5 h, 2.5 × 10^5^ (PFU/mL) of ZIKV was incubated in 1% BSA/PBS at varying pH for 1.5 h. Next, human anti-ZIKV IgG or Fab was three-fold diluted from the indicated concentration and incubated for another 1.5 h. For indirect ELISA, 2 µg/mL of recombinant ZIKV Envelope protein (rZIKV E) was coated on ELISA plates at 4 °C overnight. After blocking with 3% BSA/PBS for 1.5 h, human anti-ZIKV IgG or Fab was three-fold diluted from the indicated concentration and incubated for another 1.5 h. Subsequently, alkaline phosphatase (AP) conjugated goat anti-human IgG F(ab’)2 specific antibody (Jackson Immuno Research Laboratories, West Grove, PA, USA) was added and incubated at room temperature for 1 h, followed by color development and visualization using phosphatase substrate (Sigma-Aldrich, Waltham, MA, USA). The absorbance was read at OD 405 nm by an ELISA reader (BioTek Instruments, Winooski, VT, USA).

### 2.5. Shotgun Mutagenesis Epitope Mapping

Epitope mapping by alanine-scanning mutagenesis [48] was performed essentially as described previously [49]. A ZIKV prM-E protein expression construct (strain SPH2015) was subjected to high-throughput alanine scanning mutagenesis to generate a comprehensive mutation library. Each residue within prM-E was mutated to alanine, except the alanines which were mutated to serines. In total, 672 ZIKV prM-E mutants were generated and expressed transiently in HEK293T cells for 22 h in 384-well plates. Cells expressing the protein mutants were fixed in 4% (*v*/*v*) paraformaldehyde (Electron Microscopy Sciences, Hatfield, PA, USA), and permeabilized with 0.1% (*w*/*v*) saponin (Sigma-Aldrich) in PBS plus calcium and magnesium (PBS++). For mapping, mAb ADI-30056 was digested, then cells were sequentially incubated with 1.0 μg/mL purified Fab and 3.75 μg/mL of AlexaFluor488-conjugated secondary antibody (Jackson Immuno Research Laboratories, PA, USA), which were diluted in PBS, 10% normal goat serum (Sigma), and 0.1% saponin. Cells were washed three times with PBS++/0.1% saponin, and twice with PBS, then mean cellular fluorescence was recorded using a high-throughput flow cytometer (HTFC, Intellicyt, Albuquerque, NM, USA). Antibody reactivity against each prM-E mutant, relative to the wild-type protein, was calculated by subtracting the signal from mock-transfected controls and normalized to the signal from wild-type prM-E-transfected controls.

### 2.6. Cryo-EM Reconstruction and Structure Refinement of ZIKV-ADI-Fab30056 Complex

ZIKV strain H/PF/2013 was purified according to previously described protocol [29]. Fab fragments were generated from ADI-30056 antibody using a Pierce^TM^ Fab preparation kit. Purified ZIKV preparation in complex with ADI-Fab30056 was prepared by adding Fab-30056 to a final concentration of 2.5 µM at pH 8.0. This complex was flash-frozen on lacey carbon EM grids.

Micrographs were collected using a dose of 30.0 e^−^/Å^2^ on an FEI Titan Krios electron microscope equipped with a Gatan K2 Summit detector using a nominal magnification of 81,000 in the “super-resolution” mode, resulting in a pixel size of 0.86 Å. A total of 2135 micrographs were collected, corrected for beam induced sample motion using MotionCor2 [50] and Contrast Transfer Function (CTF) parameters estimated using CTFFIND4 [51]. Template-based picking by FindEM2 as implemented in Appion [52] was used to box out 165,789 particles. Non-reference, 2D classification was performed with Relion to select 22,330 particles [53]. Initial model generation, orientation and center refinement of selected particles (13.5% of the total boxed out particles) from eight-fold binned data was performed using the program jspr [54,55]. Further high-resolution icosahedral refinement for corrections for astigmatism, defocus, elliptical distortion and magnification with unbinned data using soft masks generated a cryo-EM map with an average resolution of 4.0 Å using the 0.143 Fourier shell correlation criterion [56].

A homology model of ADI-Fab30056 was generated using the PIGS software [57]. Density-guided real-space refinement of the ZIKV (PDB ID: 6CO8) and ADI-Fab30056 model into the 4.0 Å cryo-EM map was performed using the programs *COOT* [58] and *PHENIX.REFINE* [59]. The refinement converged to a real-space correlation coefficient of about 0.8 between the observed and calculated electron potential maps in the vicinity of the fitted model. The data statistics for the whole virus map and the refinement statistics for the icosahedral asymmetric unit are summarized in Table 1.

### 2.7. Multiple Sequence Alignment, 3D Superposition, Residue Contact Analysis and Calculation of Surface Properties

Consensus sequences of ZIKV, DENV, YFV, WNV and JEV were searched and aligned using Virus Pathogen Database and Analysis Resource (https://www.viprbrc.org). All structure superpositions were performed using the program HOMOLOGY [60]. Interactions between the residues of E proteins and ADI-Fab30056 were calculated using the CCP4 program *CONTACT* [61]. Surface residues were plotted using the program *RIVEM* [62]. All figures were prepared using the program *CHIMERA* [63].

### 2.8. Data Availability

Cryo-EM reconstructed maps of ZIKV-ADI-Fab30056 complex at 4.0 Å were deposited with the EMDataBank (EMD-22818). The refined coordinates of ZIKV-ADI-Fab30056 complex at 4.0 Å have been deposited with the Protein Data Bank (accession code PDB 7KCR).

## 3. Results

### 3.1. Structure of ZIKV in Complex with ADI-Fab30056

Complex of purified ZIKV (H/PF/2013) with ADI-Fab30056 was prepared at pH 8.0 and incubated for half an hour on ice before flash freezing on lacey carbon grids for cryo-EM data collection. Micrographs of uncomplexed ZIKV showed smooth surfaced spherical particles whereas ZIKV particles in complex with ADI-Fab30056 were mostly spiky. Higher Fab to E ratios distorted the particles. About 22,300 particles of ADI-Fab30056 complex were selected from the micrographs using non-reference two-dimensional (2D) classification (Appendix A). High resolution icosahedral refinement of the orientation, center, astigmatism and magnification of the 2D projections were performed using the jspr program [54]. This generated a cryo-EM map at an average resolution of 4.0 Å for ADI-Fab30056 according to the 0.143 Fourier shell correlation (FSC) criterion (Figure 2 and Appendix A).

The electron potential map of the ADI-Fab30056-ZIKV complex contoured at 3σ displayed density for one Fab molecule per icosahedral asymmetric unit (Figure 2a,b and Appendix A). Fabs bind to the E proteins near to the icosahedral two-fold (E2), three-fold (E3) or five-fold (E5) axes (Figure 1a) are labelled as 2f, 3f and 5f, respectively. The stoichiometry and occupancy of Ab binding depends on the accessibility of the epitope on the virus surface. In the ADI-Fab30056-ZIKV complex, the observed Fab molecule bound at the 3f site corresponds to two Fab molecules per raft (Figure 2). No Fab binding was observed at the 5f sites and very weak Fab density was observed at the 2f sites on a map contoured at 1.5σ. The occupancy of the Fabs at the 2f sites was negligible compared to occupancy at the 3f sites as the density at the 2f sites was not visible in a map contoured at 3.0σ. The 3f Fab binding site in ADI-Fab30056-ZIKV complex structure is similar to the 3f Fab binding site of a previously determined structure of ZIKV in complex with ZIKV-117 Fab at a resolution of 6.2 Å [45] (Appendix A). However, the ZIKV-117 complex structure displayed an additional Fab binding site at the 2f site with overlapping densities (Appendix A).

The cryo-EM map of the ZIKV-ADI-Fab30056 complex has weak overlapping densities at 1.5σ between the variable regions of the bound Fabs at the 3f, 2f, 2f’ and 3f’ sites, where the 2f’ and 3f’ sites are on an adjacent icosahedral asymmetric unit (Figure 2 and Appendix A). Therefore, all four sites (3f, 2f, 2f’ and 3f’) on the raft cannot be occupied at the same time because of steric hindrance. Under conditions with saturated Fab concentrations, three Fab binding states are possible: (a) 3f–3f’, (b) 3f–2f’ and (c) 3f’–2f (Appendix A). In (a), all 60 3f sites can be occupied with no bound Fabs at the 2f sites, whereas in (b) or (c) 30 3f sites and 30 2f sites can be simultaneously occupied by Fabs. The 30 rafts forming the virus particle can have different percentages of (a), (b) or (c). For example, the structure of ZIKV in complex with the nAb ZIKV-117 showed that the occupancy of the Fab at the 3f site was 69% in contrast to 38% occupancy of the Fab at the 2f site [45]. Here, the structure of ADI-Fab30056-ZIKV complex has scenario (a) as the predominant Fab binding state (Figure 3). Because of the very weak density at the 2f site, those occupancies were not calculated.

Real-space refinement of the ZIKV model (PDB ID: 6CO8) and the Fab model (generated using the PIGS software [57]) against the 4.0 Å cryo-EM map converged to a real-space correlation coefficient of 0.8 between the observed and calculated electron potential maps. The data and refinement statistics are summarized in Table 1. In contrast to the structure of ZIKV in complex with the nAb ZIKV-117, which was determined to a resolution of 6.2 Å, the main chain and side chain densities of the complementarity-determining region (CDR) loops were clearly visible in the structure of ZIKV-ADI-Fab30056 complex. Therefore, it was possible to unambiguously model the Fab heavy and light chains into the cryo-EM map (Figure 2D) and also to map the epitope residues and their interactions with the Fab molecule with great precision.

### 3.2. ADI-Fab30056 Interactions with ZIKV and ZIKV Specific Neutralization

The quaternary epitope of ADI-Fab30056 is formed by the intra-dimer (E3–E5) and inter-dimer interface (E2–E5) residues (Table 2). The bound Fab at the 3f site interacts with the loops on the E protein domain II (E-DII). The ZIKV-Fab complex is stabilized by the interactions between (1) the Fab-variable light (Fab-VL) complementarity-determining region (VL-CDR) loops 1–3 with the intra-dimer interface residues (E3–E5) and (2) the Fab-VH CDR loops 1–3 with the inter-dimer interface residues (E2–E3) (Figure 4; Table 2). The interaction surface area between the E proteins E5 and E2 and the Fab-VH loops are 634.7 Å^2^ and 148.2 Å^2^, respectively. The interaction surface area between the E proteins E5 and E3 and the Fab-VL loops are 296.7 Å^2^ and 133.8 Å^2^, respectively. Therefore, although it seems necessary to have the quaternary epitope from the arrangement of three adjacent monomers on the virus surface for the formation of a stable complex, most of the stabilizing interactions occur between the E protein E5 and Fab-VH and Fab-VL loops.

The quaternary structural epitope formed by the E protein inter-and intra-dimer interface is only accessible on the surface of the mature virion and is not accessible on immature particles or fusogenic particles which exist at low pH. ADI-30056 might be less efficient in binding both fusogenic ZIKV particles formed at a lower pH and to partial epitope available on recombinantly expressed dimeric E protein. Therefore, we tested the binding of ADI-30056 immunoglobulin G (IgG) and Fab at pH 8.0, 7.0 and 5.5 on the whole virus and on recombinantly expressed dimeric E protein (rE) (Figure 3c–f). The binding of either ADI-30056 Fab or IgG to rE was not affected by the differing pH environments. In contrast, both Fab and IgG binding to the whole virion was reduced following exposure of viral particles to an acidic environment. However, one cannot rule out the possibility that there might be some aggregation of the virus particles at lower pH. Testing our frozen grids with native virus at a lower pH did not show any visible turbidity due to aggregation. At first glance, the differences in binding curves at different pH values suggest that binding of ADI-30056 requires a pre-pH triggered maturation state and that the ADI-30056 quaternary epitope is impacted by virion acidification. However, the small difference in the binding affinities of rE and the whole virus (Figure 3c–f) suggests that the nAb ADI-30056 binds dimeric rE. This is in agreement with what the structure suggests, where most of the interactions are centered on one of the E monomers (E5). However, on the surface of the virus, the Fab-variable Heavy Chain (Fab-VH)-CDR2 loop interactions with the E protein inter-dimer interface loops (b-c and h-i) lock the two dimers, E3-E5 and E2-E2’ (Figure 4). Therefore, the two Fabs at the 3f site (or a single Ab molecule) could span three dimers on the virus surface, locking the raft and preventing further conformational changes to form a fusogenic virus.

A road map was calculated showing the surface-exposed E protein residues and their interactions with ADI-Fab30056 (Figure 4a). The map shows that the epitopes near the inter-dimer interface are overlapping and have quasi-two-fold symmetry. Therefore, the bound Fab straddles both sides of the quasi-two-fold axis allowing only one Fab molecule to bind at a given time. Weak density was observed for Fabs bound at the 2f site because of icosahedral averaging of 3f-2f’, 3f’-2f and 3f-3f’ states, where 3f-3f’ is the predominant Fab bound state in the structure of ZIKV in complex with ADI-Fab30056 reported here (see Section 3.1). The quaternary epitope defined by the interface between and within both the dimers is required for the formation of a stable ZIKV-ADI-Fab30056 complex. Thus, the partial epitope at the 5f site consisting of only one half of the epitope (b-c and h-i loops) shows no bound Fab. A comparison of the interactions between the Fab at the 3f site and the Fab modelled at the 2f site shows a few missing interactions between the E proteins near the two-fold (E2-E2’) and the 2f Fab. This also explains the preference of the 3f site over the 2f site for Fab binding. Shotgun mutagenesis epitope mapping of ADI-30056 on a ZIKV prM/E alanine scan mutation library (see Methods) and binding assays using flow cytometry identified three alanine mutations, D67A, Q89A and K118A, that each reduced ADI-Fab30056 binding (Figure 4b,c). Consistent with this, the structure showed that these E protein residues (all on a single E monomer, E5) interact with both Fab-VH-CDR2 and Fab-VH-CDR3 loops and, therefore, stabilize the ZIKV-ADI-Fab30056 complex (Figure 4). Several geographically distributed ZIKV strains were efficiently neutralized by ADI-30056, therefore providing broad range protection against different strains of ZIKV (Appendix A).

A comparison of the ZIKV-ADI-Fab30056 complex with uncomplexed ZIKV structures shows structural differences near the i-j loop, adjacent to the fusion loop. The i-j loop is displaced towards the Fab because of its interactions with the Fab-VL-CDR3 loop. This is observed in all four positions of dimers E2-E2’ and E3-E5 with varying degrees of displacement, consistent with the different occupancies of the Fabs (Appendix A). The loop i-j is important for stabilizing the E protein dimer [30] and higher concentrations of the Fab might cause major distortions of the virus particle by disrupting the E–DII and E–DI interface interactions. No other large structural differences were observed between the E proteins of the uncomplexed and Fab-bound ZIKV structures. This suggests that the formation of a stable ZIKV-ADI-Fab30056 complex does not require any major conformational changes on the surface of the E protein because the E protein loops for Fab-binding are readily accessible.

Multiple sequence alignment of several flaviviruses from an earlier study showed that the surface exposed loops along the inter-dimer interface and the glycan loop on the mature ZIKV are the most variable regions of the E protein [30]. These loops define the hotspots to induce ZIKV-specific humoral immune response and neutralization by nAbs. The nAb ADI-30056 is specific against ZIKV because the ZIKV E protein regions interacting with the CDR loops of ADI-Fab30056 show poor sequence conservation across different flavivirus species, in particular DENV (serotypes 1–4), WNV, JEV, YFV and TBEV (Appendix A).

## 4. Discussion

### 4.1. Evolution of Potent ZIKV-Specific Antibodies

Monoclonal antibodies ADI-30056 and ZIKV-117 have comparable binding affinities and bind similar regions on ZIKV. The epitope targeted by ZIKV-117 was isolated from donors in Haiti, Brazil and Mexico, whereas ADI-30056 was isolated from donors in Colombia. The relative immuno-dominance of the epitope defined by ADI-30056 in multiple donors is most likely to be due to an abundance of these types of antibodies in the germline (native) repertoire. In the current study, the serum samples were collected from ZIKV infected donors with a pre-existing DENV infection [46]. Five months’ post-infection with ZIKV, these donors developed highly potent ZIKV specific antibodies. More than 50% of these ZIKV-specific antibodies had overlapping epitopes with ADI-30056. These nAbs displayed similar neutralization and binding affinities and also utilized the same germline sequence (Figure 5a). The four most potent antibodies from the ZIKV-specific antibody pool (ADI-29997, ADI-30000, ADI30031, ADI-30056 with IC_50_ values of 14.98, 7.71, 5.18 and 1.78 ng/mL, respectively) were compared to gain insights into the structural basis of Ab affinity maturation. Multiple sequence alignments and structure comparisons suggested converged somatic mutations of the heavy chain VH-CDR loop residues N31, R56, N57, Y59 and VL-CDR residue E27 to be critical for binding the E protein residues of the h-i and b-c loops at the inter-dimer interface (Figure 5c; Appendix A). Minor sequence differences in the CDR-loop residues are reflected in their binding affinities. For example, the VH-CDR2 loop of ADI-29997 has the residues KHT instead of R/NNK as in ADI-30000, ADI-30031, and ADI-30056, and has lower affinity to ZIKV when compared to other antibodies in this pool. Precise tailoring of the CDR-loop residues from the ZIKV-specific antibody pool by somatic hypermutation improved the paratope complementarity to the ZIKV quaternary epitope, which is in line with the evolution of ZIKV nAb molecules with higher binding affinities. Structure comparisons and sequence alignments of ZIKV and DENV strains at the inter-dimer interface show major epitope differences between the two viruses (Figure 5c,d). The most striking difference is the glycosylation of the ZIKV D67 equivalent residue, N67 in DENV. The glycosylation of N67 sterically hinders nAb access to the inter-dimer interface among the four serotypes of DENV. Therefore, an antibody similar to ADI-30056 or ZIKV-117 that locks the inter-dimer interface has not been so far observed in DENV.

### 4.2. Translational Applicability of ADI-30056 for Ab-Based Therapeutics

ADI-30056 is a highly potent ZIKV-specific nAb targeting the inter-dimer interface similar to ZIKV-117. ZIKV-117 displayed higher neutralizing activities compared to other ZIKV specific nAbs and protected mice against ZIKV infection by reducing tissue pathology, vertical transmission and mortality [49]. ZIKV-117 is being developed as a therapeutic antibody treatment against ZIKV infection by the interdisciplinary teams at Vanderbilt University Medical Center, IDBiologics and Batavia Biosciences. Another robust translation application was the intramuscular delivery of replicon RNA encoding ZIKV-117, which protected mice from ZIKV infection both as a pre-exposure prophylaxis and post-exposure therapy [64]. The structure of ADI-30056 in complex with ZIKV provides an initial model for structure based design approach. The high resolution atomic details of ZIKV-ADI-Fab30056 can be used for re-designing the Ab CDR loops. This would greatly improve the epitope-paratope complementarity and enhance the therapeutic potential of dimer cross-linking nAbs. Similar approach might be utilized to design Abs that bind other flaviviruses such as WNV and JEV at the inter-dimer interface.

## 5. Conclusions

ZIKV-specific neutralization by ADI-30056 is achieved by the binding of one Ab molecule onto the quaternary epitope spread among three adjacent dimers. This would inhibit conformational changes required for the formation of fusogenic trimers and cell entry. ADI-30056 is highly potent because, in addition to ZIKV-specific binding, it might prevent further spread of infection. Secondly, the conformational differences observed at the i-j loop near the fusion loop might destabilize the dimers, showing that higher concentrations of ADI-30056 might also prematurely destroy the E protein dimers (also observed when higher concentrations of Fab concentrations are mixed with the virus), protecting the genome of the virus. However, this would need further experimental evidence. The accessibility of other cross-reactive antibodies on the surface of ZIKV might also be limited upon binding of ADI-30056 due to steric hindrance (Appendix A). With no approved vaccine or antiviral treatment against ZIKV, the near-atomic resolution molecular analysis of ZIKV neutralization by ADI-30056 is a step further in the development of an antibody-based therapeutic.

## Figures and Tables

**Figure 1 viruses-12-01346-f001:**
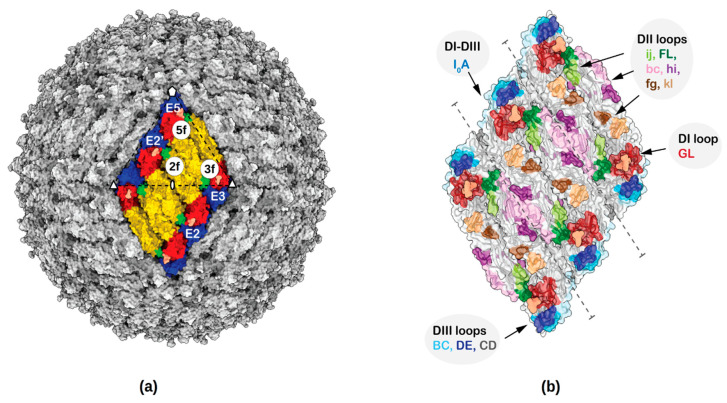
Fab binding sites and surface accessible antigenic loops of Zika virus (ZIKV). (**a**) ZIKV structure showing the herringbone pattern formed by 6 E–M heterodimers. One icosahedral asymmetric unit is identified by a black triangle. For clarity only the ectodomain of the E protein (from residues 1–400) has been shown. The Fab binding sites near the two-fold, three-fold and five-fold axes of symmetry are labelled as 2f, 3f and 5f, respectively and the E proteins near the two-fold, three-fold and five-fold axes of symmetry are labelled E2, E3 and E5, respectively. Domains E-DI, E-DII and E-DIII are colored red, yellow and blue, respectively. (**b**) Surface representation of various loops accessible to nAbs are labelled.

**Figure 2 viruses-12-01346-f002:**
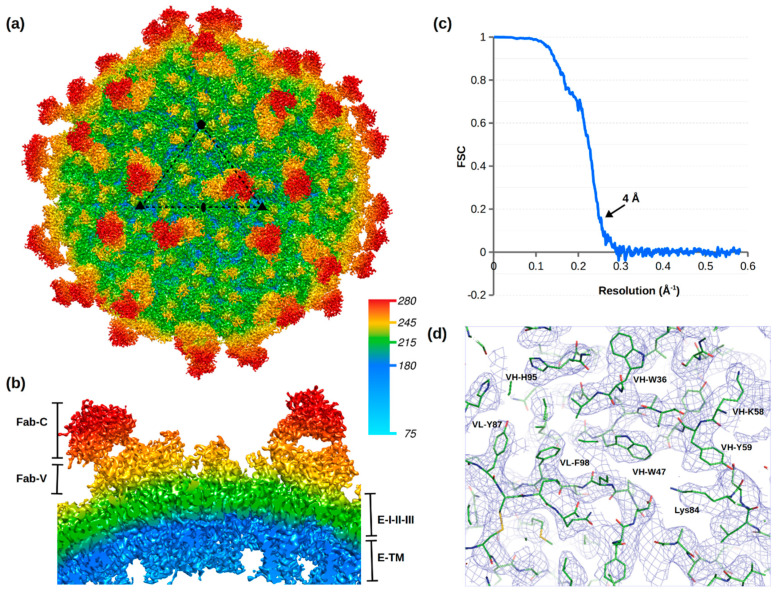
Structure of ZIKV in complex with ADI-Fab30056. (**a**) Cryo-EM map of ZIKV in complex with ADI-Fab30056 at a resolution of 4 Å viewed down an icosahedral two-fold axis contoured at 3σ. The icosahedral asymmetric unit is outlined by a black triangle. (**b**) Cross-section of the cryo-EM map showing the Fab-constant (Fab-C), Fab-variable (Fab-V), E protein domains (I-II-III) and transmembrane region. Radial map coloring is labelled with the distance from the virus center in Å. (**c**) Fourier shell coefficient (FSC) plot versus resolution. (**d**) Cryo-EM map around ADI-Fab30056 and ZIKV residues contoured at 3σ. Variable heavy (VH) and variable light (VL) residues are labelled in single letter code and E protein residues in three letter code.

**Figure 3 viruses-12-01346-f003:**
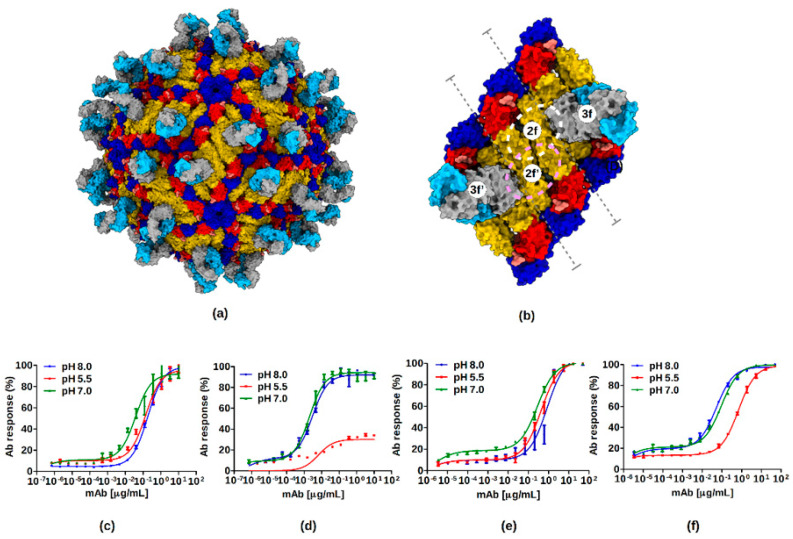
ZIKV neutralization by ADI-Fab30056. (**a**) Binding sites of ADI-Fab30056 at the 3f site on the virus surface. The three E protein domains (I, II, and III) are colored red, yellow and blue, respectively. The Fab-variable heavy chain (Fab-VH) and the Fab-variable light chains (Fab-VL) are colored in grey and cyan, respectively. (**b**) Binding sites of ADI-Fab30056 at the 3f site on an isolated raft. Binding of Fabs at the 2f/2f’-sites are outlined with white and pink dashed circles, respectively. The inter-dimer interface is marked by dashed grey lines. (**c**–**f**) Binding curves of anti-ZIKV ADI-30056 Fab and IgG on whole virion and rZIKV E at different pH values (8.0, 5.5 and 7.0) analyzed using ELISA. The binding of Fab to (**c**) rZIKV E and (**d**) whole Zika virion, strain Paraiba. The binding of IgG to (**e**) rZIKV E and (**f**) whole Zika virion, strain Paraiba. All data is presented as the mean ± standard error from at least two independent experiments.

**Figure 4 viruses-12-01346-f004:**
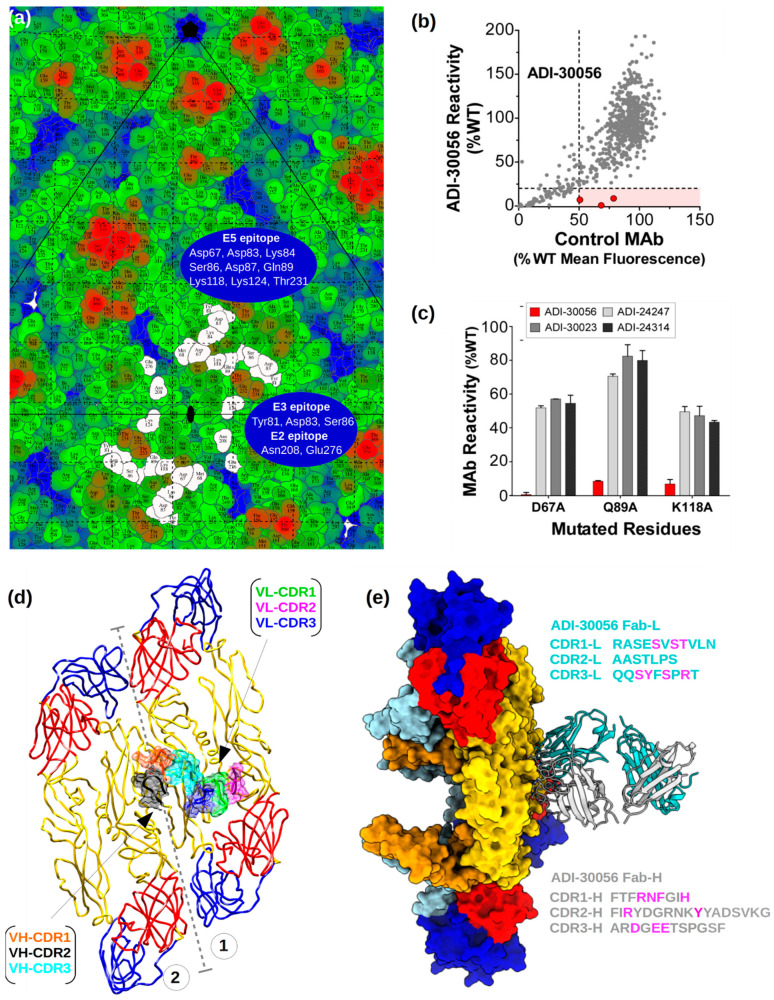
ADI-Fab30056 epitope mapping. (**a**) Road map of ZIKV E protein residues. The footprint of the E protein residues interacting with ADI-Fab30056 are represented in white at the 3f site. The E protein residues are colored from blue to red through green according to their distance from the virus center. The E2, E3 and E5 epitope residues are labelled on the roadmap. (**b**,**c**) Shotgun mutagenesis epitope mapping of ADI-30056. ADI-30056 was epitope mapped by screening on a ZIKV prM/E alanine scan mutation library expressed in human embryonic kidney (HEK) 293T cells, with binding assayed by flow cytometry. (**b**) Cells expressing the ZIKV prM/E mutation library were tested for immunoreactivity with a Fab version of 30056, and also by a control mAb. A comparison of the reactivity’s for the two Abs across the entire library identified three critical clones (shown in red) that showed reduced binding for 30056 Fab (<20% of binding to wild-type ZIKV prM/E), but a high level of binding to the control mAb. Reactivities are expressed as a percentage of Ab bound to wild-type ZIKV prM-E. (**c**) The three Domain II mutants identified in panel A (D67A, Q89A and K118A) each reduced ADI-Fab30056 binding (red bars) but did not affect binding of control mAbs 24247, 30023, and 24314 (gray and black bars). Bars represent the mean binding and range (half of the maximum minus minimum values) of at least two replicate data points. (**d**) Top view of the interactions between ADI-Fab30056 VH-CDR and VL-CDR loops and ZIKV E protein loops at the inter-dimer and intra-dimer interface are shown. The VH and VL CDR loops 1, 2, and 3 are colored orange, black, cyan, green, magenta and blue, respectively. E protein ectodomain coloring is as in Figure 1. The inter-dimer interface is marked by dashed grey lines. (**e**) Side view of the ADI-Fab30056 interaction with the E proteins in the asymmetric unit. The Fab-variable heavy chain (Fab-VH) and the Fab-variable light chains (Fab-VL) are colored in grey and cyan, respectively. The residues of the CDR loops critical for epitope recognition are colored in magenta.

**Figure 5 viruses-12-01346-f005:**
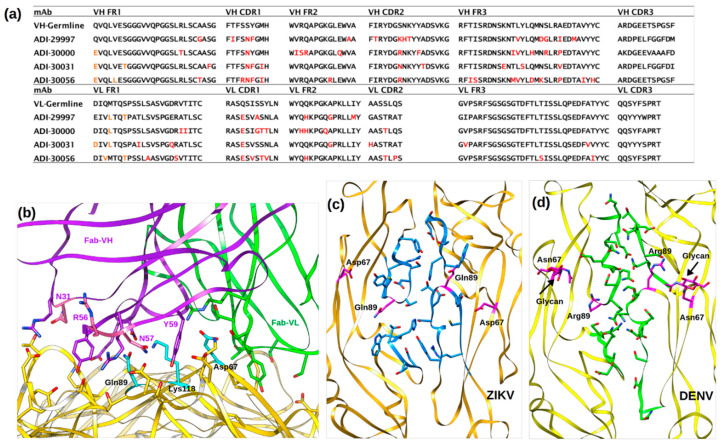
Neutralization by ZIKV specific antibodies. (**a**) Sequence alignment of ADI-30056 VH and VL regions with other ZIKV specific potent mAbs that utilized the same VH germline (VH3-30). Somatic mutations are shown in red. (**b**) Molecular interactions of ADI-Fab30056 CDR loop residues with ZIKV D-II residues. E protein, Fab-VH and Fab-VL are rendered yellow, purple and green, respectively, and the residues important for ZIKV-Fab interactions are shown as sticks. (**c**,**d**) Inter-dimer interface residues (blue sticks in ZIKV and green sticks in DENV) observed in ZIKV and DENV. Both ZIKV and DENV are rendered as ribbons and shown in yellow. The ZIKV residues Q89 and D67 as well as the DENV residues R89 and N67-glycan are shown as magenta sticks.

**Table 1 viruses-12-01346-t001:** Data collection and refinement statistics.

Data Statistics	
Microscope	FEI Titan Krios
Magnification (×)	81,000
Voltage (kV)	300
Detector	Gatan K2 Summit
Micrograph collection mode	Super-resolution
Total dose (e-/Å^2^)	30
Defocus range (m)	1.0–2.5
Pixel size (Å)	0.86
Symmetry imposed	Icosahedral
Map resolution (FSC 0.143; Å)	4.0
Map sharpening B-factor (Å^2^)	−200.86
**Refinement statistics**	
Real-space correlation coefficient	0.8
Root-mean-square deviation (r.m.s.d.) bond lengths (Å)	0.02
r.m.s.d bond angles in degrees (º)	1.25
Clash score	13
Poor rotamers (%)	4.5%
Ramachandran plot	
Favored (%)	88.4
Allowed (%)	11.6
Outliers (%)	0.0

**Table 2 viruses-12-01346-t002:** Interactions of ADI-30056-Fab at the 3f site with E proteins at the two-fold (E2), three-fold (E3) and five-fold (E5) axes.

E5	Fab Heavy (H) or Light (L) Chains	Distance (Å)
Asp83 (b-c loop)	H:Y59 (VH-CDR2)	2.86
Ser 86 (b-c loop)	H:Y59 (VH-CDR2)	3.55
Asp87 (b-c loop)	H:R52 (VH-CDR2)	2.66 (salt bridge)
Thr88 (b-c loop)	H:R52 (VH-CDR2)	3.28
Gln89 (b-c loop) **	H:E101 (VH-CDR3)	2.94
Lys118 (strand-d) **	H:E101 (VH-CDR3)	3.46 (salt bridge)
Lys124 (d-e loop)	H:R52 (VH-CDR2)	2.88
Lys124 (d-e loop)	H:E102 (VH-CDR3)	2.88 (salt bridge)
Thr231 (h-i loop)	H:N31 (VH-CDR1)	2.77

Asp67 (strand-b) **	L:S91 (VL-CDR3)	2.71
Asp67 (strand-b)	L:R96 (VL-CDR3)	3.61 (salt bridge)
Met68 (strand-b)	L:S94 (VL-CDR3)	3.39
Met68 (strand-b)	L:Y92 (VL-CDR3)	2.75
Lys84 (b-c loop)	L:S94 (VL-CDR3)	2.53
Val255 (i-j loop)	L:Y92 (VL-CDR3)	3.78
**E2**	**Fab Heavy (H) or Light (L) chains**	**Distance (Å)**
Tyr81 (b-c loop)	H:Y53 (VH-CDR2)	3.76
Asp83 (b-c loop)	H:R30 (VH-CDR1)	2.40 (salt bridge)
Ser86 (b-c loop)	H:Y53 (VH-CDR2)	2.27
Ser86 (b-c loop)	H:R52 (VH-CDR2)	2.75
**E3**	**Fab Heavy (H) or Light (L) chains**	**Distance (Å)**
Asn208 (f-g loop)	L:T31 (VL-CDR1)	3.38
Asn208 (f-g loop)	L:S67 (VL-FR3)	3.59
Glu276 (strand-K)	L:S28 (VL-CDR1)	3.45

** The residues that significantly reduce ADI-30056 binding when mutated to alanine.

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
