# Peer review of "Structural Basis of Zika Virus Specific Neutralization in Subsequent Flavivirus Infections"

_viruses, 2020, doi:10.3390/v12121346_

Round 1

Reviewer 1 Report

Sevvana et al. report on a high resolution structure of mature Zika virus in complex with a human monoclonal antibody from an individual infected with Zika virus and with pre-existing immune responses to Dengue virus.

The finding that the mAb binds a quaternary structural epitope on E-protein is not novel but the experimental evidence presented here is rigorous and provides additional molecular insights in the binding and neutralizing capacity of this mAb.

Just one minor remark: the authors could expand the discussion section with a paragraph on the translational applicability of this mAb for Ab-based therapies.

Author Response

We thank the reviewers for their supportive comments.

Response to reviewer 1:

Sevvana et al. report on a high resolution structure of mature Zika virus in complex with a human monoclonal antibody from an individual infected with Zika virus and with pre-existing immune responses to Dengue virus.

The finding that the mAb binds a quaternary structural epitope on E-protein is not novel but the experimental evidence presented here is rigorous and provides additional molecular insights in the binding and neutralizing capacity of this mAb.

  1. Just one minor remark: the authors could expand the discussion section with a paragraph on the translational applicability of this mAb for Ab-based therapies.
  • Section 4.2 discussing the translational applicability of ADI-30056 has been added as suggested by the reviewer.

Reviewer 2 Report

This paper described a cryo-EM reconstruction of the mature ZIKV in complex with ADI-19, a ZIKV-specific human monoclonal antibody (hMAb) isolated from a ZIKV infected donor with a prior dengue virus infection. Overall, the paper is well-written, nicely referenced, and properly controlled. This is solid work that deserves publication in Viruses. I have a couple of suggestions that could strengthen this work. I ask the authors to consider my suggestions:

1 – Figure 2a shows a cryo-EM of the Zika capsid decorated with ADI-19. The color-coding (based on the distance from the particle's center) makes it a bit difficult to discern the bound antibodies. Likewise, in the text, the authors refer to some partially occupied vs fully occupied antibodies present in the icosahedrally averaged reconstruction. I believe it would be useful to see all densities for antibodies in the reconstruction cut out of the overall density, fit to their pseudo-atomic models (after real-space refinement), and illustrated at the same contour in a dedicated figure (in the main text or supplementary). This would greatly help the reader follow the description of structural data.

2 – The authors do not investigate why the occupancy of the Fabs at the 2f sites is negligible compared to occupancy at the 3f sites. Can this point be discussed a bit more? Is it possible the weaker density reflects structural heterogeneity at the binding site and/or deviation from the imposed icosahedral symmetry? Have the authors attempted to carry out symmetry expansion and focus classification of the density at the 2f sites?

Overall, this is a strong paper that deserves publication in Viruses

Author Response

Response to reviewer 2:

This paper described a cryo-EM reconstruction of the mature ZIKV in complex with ADI-30056, a ZIKV-specific human monoclonal antibody (hMAb) isolated from a ZIKV infected donor with a prior dengue virus infection. Overall, the paper is well-written, nicely referenced, and properly controlled. This is solid work that deserves publication in Viruses. I have a couple of suggestions that could strengthen this work. I ask the authors to consider my suggestions:

1 – Figure 2a shows a cryo-EM structure of the Zika capsid decorated with ADI-30056. The color-coding (based on the distance from the particle's center) makes it a bit difficult to discern the bound antibodies. Likewise, in the text, the authors refer to some partially occupied vs fully occupied antibodies present in the icosahedrally averaged reconstruction. I believe it would be useful to see all densities for antibodies in the reconstruction cut out of the overall density, fit to their pseudo-atomic models (after real-space refinement), and illustrated at the same contour in a dedicated figure (in the main text or supplementary). This would greatly help the reader follow the description of structural data.

  • We agree with reviewer 2 that the color coding makes it difficult to discern the bound antibodies. Therefore, we provide figure 2b showing the cross-section of the map with the bound antibodies color coded similar to figure 2a.
  • We have also added supplementary figure S2c to visualize the cryo-EM map contoured at 3s around the envelope protein and the bound ADI-30056 at the 3f site. The density near the 2f site is very weak and negligible as described in the text (L235-238).

2 – The authors do not investigate why the occupancy of the Fabs at the 2f sites is negligible compared to occupancy at the 3f sites. Can this point be discussed a bit more? Is it possible the weaker density reflects structural heterogeneity at the binding site and/or deviation from the imposed icosahedral symmetry? Have the authors attempted to carry out symmetry expansion and focus classification of the density at the 2f sites?

We have discussed the occupancy of Fabs at the 3f and 2f sites at: 243-253, L335-340 and L343-355. We have added an additional sentence to clarify this further (L337-340).

“Weak density was observed for Fabs bound at the 2f site because of icosahedral averaging of 3f-2f’, 3f’-2f and 3f-3f’ states, where 3f-3f’ is the predominant Fab bound state in the structure of ZIKV in complex with ADI-Fab30056 reported here (please also see section 3.1).”

The weaker density does reflect structural heterogeneity and deviation from icosahedral symmetry. We have not performed localized reconstruction and 3D classification as the density at the 2f sites was very weak. We would have done it had the structure been similar to ZIKV-117 with ~40% occupancy of the Fabs at the 2f sites and ~60% at the 3f sites. We have described this in the text that we have not calculated the occupancies because of negligible density at the 2f sites. However, we discuss about the possibility of binding at 3f-2f’ sites.